# The anti-cancer drugs curaxins target spatial genome organization

Omar L. Kantidze [1], Artem V. Luzhin [1], Ekaterina V. Nizovtseva[2], Alfiya Safina[3], Maria E. Valieva[4], Arkadiy K. Golov[1], Artem K. Velichko[1], Alexander V. Lyubitelev[4], Alexey V. Feofanov[4,5], Katerina V. Gurova[3], Vasily M. Studitsky[2,4] & Sergey V. Razin[1,4]

Recently we characterized a class of anti-cancer agents (curaxins) that disturbs DNA/histone interactions within nucleosomes. Here, using a combination of genomic and in vitro approaches, we demonstrate that curaxins strongly affect spatial genome organization and compromise enhancer-promoter communication, which is necessary for the expression of several oncogenes, including *MYC*. We further show that curaxins selectively inhibit enhancer-regulated transcription of chromatinized templates in cell-free conditions. Genomic studies also suggest that curaxins induce partial depletion of CTCF from its binding sites, which contributes to the observed changes in genome topology. Thus, curaxins can be classified as epigenetic drugs that target the 3D genome organization.

[1] Institute of Gene Biology RAS, 34/5 Vavilov Str., 119334 Moscow, Russia. [2] Cancer Epigenetics Program, Fox Chase Cancer Center, 333 Cottman Ave., Philadelphia, PA 19422, USA. [3] Department of Cell Stress Biology, Roswell Park Comprehensive Cancer Center, Elm and Carlton St, Buffalo, NY 14263, USA. [4] Biology Faculty, Lomonosov Moscow State University, 1 Leninskie Gory, 119992 Moscow, Russia. [5] Shemyakin-Ovchinnikov Institute of Bioorganic Chemistry, Russian Academy of Sciences, 16/10 Miklukho-Maklaya Str., 117997 Moscow, Russia. These authors contributed equally: Omar L. Kantidze, Artem V. Luzhin, Ekaterina V. Nizovtseva, Alfiya Safina. Correspondence and requests for materials should be addressed to K.V.G. (email: katerina. gurova@roswellpark.org) or to V.M.S. (email: vasily.studitsky@fccc.edu) or to S.V.R. (email: sergey.v.razin@usa.net)

Three-dimensional (3D) genome organization is now considered an additional level of the epigenetic regulation of gene expression. Techniques exploiting the proximity ligation principle and deep sequencing have demonstrated that the genome is partitioned into spatially segregated transcriptionally more active A and inactive B compartments[1]. On a sub-megabase level, chromatin fiber is folded into topologically associating domains (TADs) that are composed of smaller contact domains, most of which are chromatin loops[2–4]. In mammalian cells, these local-scale structures are maintained by the architectural proteins CTCF and cohesin and some transcription factors[5]. Chromatin loops are thought to bring gene promoters to their cognate cis-regulatory elements within the same TAD[6]. In higher eukaryotes, the number of enhancers significantly exceeds the number of known genes[7], and expression of a single gene may be controlled by various combinations of the enhancers and silencers[8]. In the course of gene activation, the juxtaposition of enhancer and target promoter creates an active phase-separated condensate[9–11], which is important for transcriptional activation of genes regulated by super-enhancers (SEs)[9].

Enhancer- and SE-mediated activation of oncogenes often underlies neoplastic cell transformation[12–14]. Therefore, inhibition of cancer-specific enhancer activity is a current direction in anti-cancer drug development[15–17]. Indeed, pharmacological inhibition of transcriptional activators that are involved in SE function leads to selective downregulation of oncogene expression and cancer cell death[18–20]. One may assume that another strategy to affect SE-driven oncogene transcription is the disruption of long-distance interactions between enhancers/SEs and target promoters. Here, we provide evidence that such an approach can be exploited in cancer treatment.

We recently discovered a class of anti-cancer agents (curaxins) that suppress the transcription of oncogenes and affect chromatin structure[21]. The lead curaxin, CBL0137, has exhibited significant efficacy in preclinical cancer models[22–26]. Although curaxins intercalate into DNA, they do not induce any detectable DNA damage, which distinguishes them from many other DNA-binding small molecule compounds[27,28]. Depending on the concentration, curaxins can induce unwrapping of DNA from nucleosomes and stimulate partial histone eviction and B- to Z-DNA transitions in vivo[27]. These effects are likely due to increased negative supercoiling.

In this study, we demonstrate that the effects that curaxins exert on the physical properties of DNA and chromatin fiber can prevent efficient long-distance enhancer-promoter communication (EPC) in vitro and in vivo. Using an in vitro approach to measure the EPC rate, we show that curaxins prevent the looping of either naked or chromatinized DNA templates. In vivo, the Hi-C technique revealed that curaxins strongly affect the spatial organization of the genome. Specifically, curaxins compromise TADs and disrupt chromatin loops to alter EPC in living cells. These effects are mediated, at least in part, by curaxin-induced depletion or dissociation of CTCF from its binding sites. Consistent with these data, curaxin CBL0137 effectively inhibits transcription of enhancer-regulated oncogenes in tumor cells. MYC family genes are among the most sensitive to this drug. Therefore, curaxins can be classified as epigenetic drugs that target 3D genome organization.

## Results

### CBL0137 suppresses enhancer-controlled transcription. 
To better understand the mechanisms of curaxin CBL0137 toxicity in cancer cells, we evaluated whether CBL0137 treatment preferentially affected the expression of genes important for neoplastic phenotype. Expression of wild-type or translocated MYC family genes (c-MYC, NMYC, and LMYC[29]) is highly suppressed by CBL0137 at both the mRNA and protein level in various cell lines (Fig. 1a, b). The effect was not observed in transient transfection experiments with a reporter gene expressed under the control of a MYC minimal promoter alone or supplemented with an enhancer (Fig. 1c). Thus, the effect of CBL0137 on MYC expression is dependent on genomic context. In a control experiment, CBL0137 strongly suppressed the activity of an NF-κB-dependent reporter (Fig. 1c), in agreement with the previously published observations[21].

To identify genes that are inhibited by CBL0137 similarly to MYC, we analyzed the effect of CBL0137 on the gene expression profiles of two human tumor cell lines, namely multiple myeloma MM1.S and fibrosarcoma HT1080 using microarray hybridization and nascent RNA-sequencing[28]. In MM1.S cells, MYC was one of the strongly inhibited genes following the 6-h CBL0137 treatment (Fig. 1d). Moreover, all genes regulated by enhancers (HT1080) or SEs (MM1.S) were strongly inhibited by CBL0137 at lower concentrations compared to genes lacking enhancers (Fig. 1e, f). Taking into account the inability of CBL0137 to suppress enhancer activity in transient transfection experiments when enhancers were placed close to the promoter (Fig. 1c), the strong effect of CBL0137 on gene transcription controlled by enhancers in living cells suggests that it is not the activity of the enhancer per se but long-distance EPC that is affected by CBL0137.

### CBL0137 suppresses enhancer-promoter communication. 
To further evaluate the possibility that curaxins affect EPC, we used a previously developed model system to quantify the rate of EPC using a chromatinized template[30]. The essential part of this system is the model construct composed of the prokaryotic glnAp2 promoter and NtrC-dependent enhancer separated by an array of regularly spaced nucleosomes assembled on a 2.5-kb DNA fragment containing repeating 147-bp high-affinity 601 nucleosome positioning sequences (NPS) with 30-bp spacers between them (Fig. 2a)[30]. This array spontaneously forms a chromatin fiber in vitro[31,32]. A saturated chromatin array contains nucleosomes formed predominantly on NPS, but not on the enhancer or promoter sequences (Supplementary Fig. 1). To analyze the effect of CBL0137 on EPC mediated by the looping of the intervening segment of a chromatin fiber, the test constructs were transcribed in the presence or absence of different concentrations of CBL0137, and the rate of EPC was measured as described[30]. Briefly, the enhancer is activated by the NtrC protein complex, which is phosphorylated by NtrB protein kinase[33]. The phosphorylated enhancer-bound NtrC stimulates conversion of the inactive, closed complex of the RNA polymerase holoenzyme to the open, functionally active initiation complex (Fig. 2b). The intervening DNA or chromatin should be transiently looped out to allow interaction of the enhancer with the RNA polymerase holoenzyme. Loop formation is the rate-limiting step in this process. Thus, the amounts of the produced transcript are directly proportional to the EPC rate[34]. Although assembly of chromatin facilitates EPC[35], EPC can be observed and also quantified on linear, histone-free DNA.

We found that addition of CBL0137 (1–2.5 μM) caused a strong decrease in the transcript yield on both the chromatinized and histone-free DNA model construct (Fig. 2c, d). This effect could be explained by (1) inhibition of distant EPC, or (2) inhibition of the enhancer or transcription by the drug. The latter possibility was evaluated using a DNA template containing the enhancer and promoter in close proximity to each other. The yield of the transcription product on this template was minimally affected by CBL0137 (Fig. 2e), suggesting that CBL0137 does not

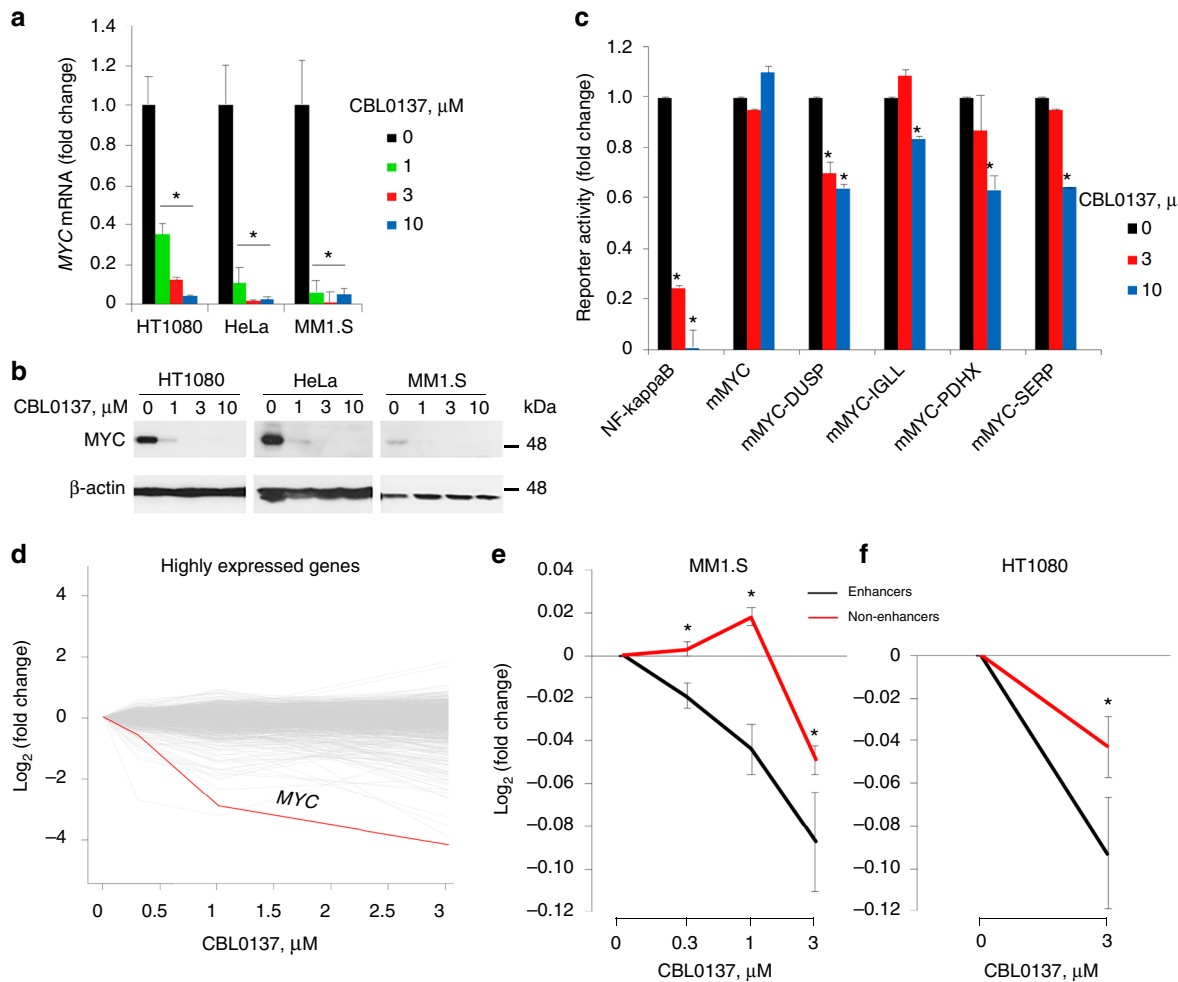

**Fig. 1** Effect of curaxin CBL0137 on transcription in cells. **a**, **b** Reduction of MYC protein and mRNA levels in HT1080, HeLa, and MM1.S cells treated with different concentrations of CBL0137 for 6 h assessed by RT-qPCR (**a**) or immunoblotting (**b**). **c** Effect of CBL0137 treatment on the activity of reporter constructs transfected into MM1.S cells and regulated by the NF-κB response element, minimal MYC promoter (mMYC) and mMYC together with enhancer elements from the indicated genes (see Methods for details). **d**–**f** Effect of CBL0137 on the level of transcripts in MM1.S (**d**, **e**) and HT1080 (**f**) cells treated with different concentrations of CBL0137 for 6 h, assessed using microarray hybridization (MM1.S cells) or nascent RNA-seq (HT1080 cells). Error bars represent the s.e.m. for three replicates. For each group of genes, the mean log FCs and standard errors of gene expression after treatment are calculated. Source data of Fig. 1a–e are provided in a Source Data file

inhibit the transcriptional machinery. It was possible that CBL0137 affected EPC in chromatin by disrupting nucleosome structure or affecting the structure and dynamics of the linker DNA supporting efficient EPC[36]. We analyzed the nucleosomal arrays using micrococcal nuclease digestion and found that CBL0137 did not affect the nucleosome structure at the concentrations that strongly affected EPC (Supplementary Fig. 2). We also used a mononucleosomal template containing 40-bp DNA linkers labeled with donor-acceptor pair fluorophores to study the effect of CBL0137 on linker internucleosomal DNA by single particle Förster resonance energy transfer (spFRET) microscopy (Fig. 2f). The proximity ratio $E_{PR}$, which is directly related to the FRET efficiency, was calculated from the fluorescence intensities of donor and acceptor fluorophores for each measured nucleosome. In agreement with previous studies[37], the frequency distribution of the nucleosomes by $E_{PR}$ value was characterized by the presence of two states with maxima $E_{PR} = 0.04 \pm 0.02$ (open linkers, donor and acceptor are far from each other) and $E_{PR} = 0.47 \pm 0.04$ (closed linkers, donor and acceptor are in close proximity), which correspond to $27 \pm 2$ and $73 \pm 2\%$ of nucleosomes, respectively (Fig. 2f). Free labeled DNA was characterized by a single state with $E_{PR} = 0.02 \pm 0.01$. Addition of

CBL0137 did not affect this state (Supplementary Fig. 3). In the presence of CBL0137, the fraction of the nucleosomes with a larger distance between the linkers increased from $27 \pm 2\%$ to $39 \pm 2\%$ (Fig. 2f and Supplementary Fig. 4), suggesting that CBL0137 changed the structure and dynamic properties of the linker DNA in the chromatin, which likely affected EPC in our model system by hampering chromatin/DNA loop formation.

**CBL0137 affects 3D genome organization in living cells**. To find out whether curaxin globally affects EPC in living cells, we performed Hi-C analysis[1] and generated chromatin interaction maps for HT1080 cells incubated with or without 3 μM CBL0137 for 6 h. At this time point, treatment did not lead to a significant increase in the number of apoptotic cells (Supplementary Fig. 5), which is consistent with our previous study that demonstrated that apoptosis occurs 48 h after treatment with this CBL0137 dose[21]. We found that CBL0137 treatment disturbed the 3D organization of the genome (Fig. 3a, b). The TAD borders appeared partially disrupted as evidenced by an increase in inter-TAD contacts and a decrease in the contact density within TADs (Fig. 3a, lower panel). Chromatin looping was also fully or

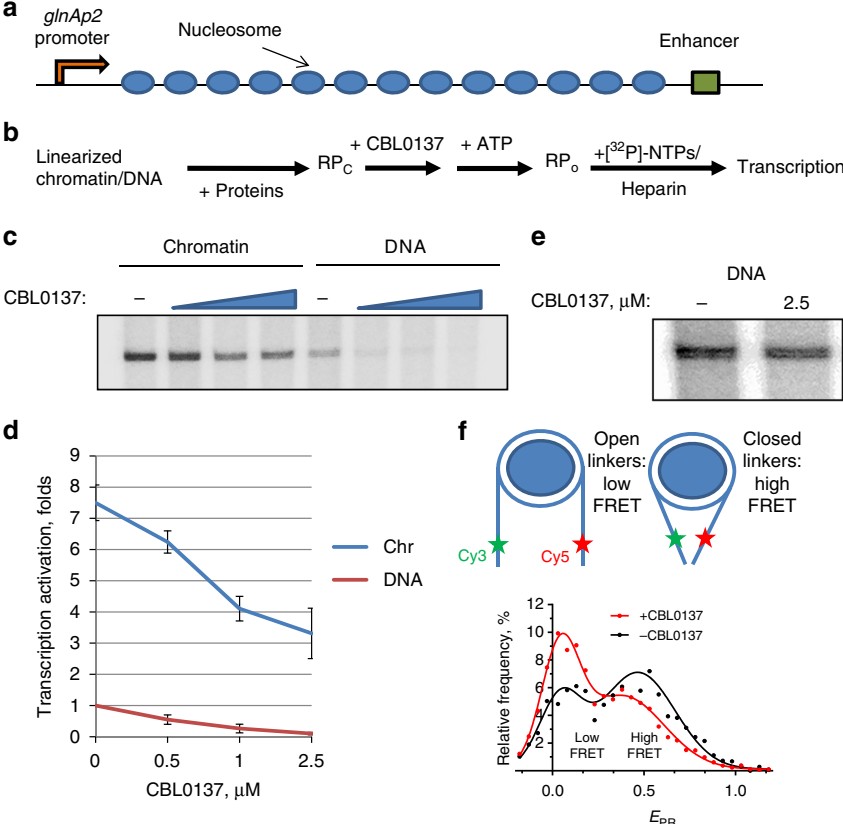

**Fig. 2** CBL0137 inhibits EPC in vitro. **a** A 13-nucleosome $601_{207 \times 13}$ array for EPC rate analysis[30, 33]. **b** Experimental approach for the EPC rate analysis on DNA and chromatin[30]. After assembly of the corresponding DNA–protein complexes at the promoter and enhancer, the array was incubated with CBL0137, and EPC was initiated. The addition of labeled rNTPs with heparin allows transcript synthesis, prevents a second round of transcription, and eliminates the nucleosomal barrier for the transcribing RNA polymerase. **c** Transcription of the array in the presence of increasing CBL0137 concentrations (0.5, 1, and 2.5 μM). Analysis of labeled transcripts by denaturing PAGE. **d** Quantitative analysis of the 176-nt transcripts shown in **c**. Error bars represent the s.d. based on four independent measurements using two different reconstitutes. **e** Transcription of the DNA array having a short distance (708 bp) between the enhancer and promoter in the presence or absence of CBL0137. **f** CBL0137 affects the conformation of linker nucleosomal DNA. Top: Fluorescently labeled nucleosomes (asterisks mark positions of Cy3 and Cy5 labels) having 40-bp extending DNA linkers were incubated with or without CBL0137 (0.5 μM). Bottom: Typical frequency distributions of nucleosomes by proximity ratio ($E_{PR}$) measured using spFRET microscopy. Experimental data (dots) were fitted with a sum of two Gaussians (solid lines). The sample sizes (n, single particle events) were: (+CBL0137) – 4310; (−CBL0137) – 2832. The mean values of $E_{PR}$ peak maxima and s.e.m. averaged over three independent experiments were: (+CBL0137) – 0.04 ± 0.01, 0.38 ± 0.04; (−CBL0137) – 0.04 ± 0.02, 0.47 ± 0.04. Source data of Fig. 2c–f are provided in a Source Data file

partially lost (Fig. 3b). Accordingly, the scaling of the distant chromatin contacts differed between control and curaxin-treated cells. CBL0137 treatment caused a decrease in the spatial interactions over distances shorter than 600 kb (i.e., mostly within TADs) and an increase in the spatial interactions over longer distances (i.e., mostly between TADs) (Fig. 3c). Furthermore, the TAD border strength defined as the ratio of inter-TAD to intra-TAD contacts averaged for all TAD pairs[38] decreased significantly after exposure to CBL0137 (Fig. 3d). The observed differences could not be explained by the presence of a few outliers because a decrease in the border strength after incubation with CBL0137 occurred for the majority of annotated TAD borders (Fig. 3e).

We next annotated A (active) and B (inactive) chromatin compartments[1] and compared the frequency of interactions between bins belonging to the different compartments in normal and CBL0137-treated cells. We found that exposure of cells to CBL0137 decreased the level of spatial segregation of the A and B chromatin compartments (Fig. 3f, g). The quantitative effect of CBL0137 on compartment segregation varied from chromosome to chromosome and may depend in part on gene density (Supplementary Fig. 6, compare chromosomes 18 and 19).

Taking into account the fact that exposure of cells to CBL0137 strongly suppressed transcription of the MYC gene, the profile of this gene contacts with the previously annotated SEs[12] was analyzed and showed that CBL0137 treatment disrupted these contacts (Fig. 4a). Next, all spatial contacts between the annotated genes and potential regulatory elements were identified using the PSYCHIC computational approach[39] (Supplementary Data 1; representative examples of annotated contacts on heatmaps are shown in Fig. 4c). In contrast to other related techniques (e.g., HiCCUPS[4] and Fit-Hi-C[40]), PSYCHIC-mediated annotation of promoter-enhancer interactions is TAD-specific[39]. Thus, the data obtained by PSYCHIC are generally more accurate and not skewed by TAD boundary elements[39]. We found that CBL0137 treatment caused a drastic redistribution of distant contacts. Only about 30% of the contacts annotated in control cells persisted after CBL0137 treatment, and a number of new spatial contacts appeared in CBL0137-treated cells (Fig. 4b and Supplementary Data 1). Given that PSYCHIC identifies intra-TAD interactions, the spatial contacts of MYC and other genes located at the TAD borders in our map escaped annotation. Among the 1585 genes with changed contact profiles, 59% were downregulated, and 41% were upregulated. Interestingly,

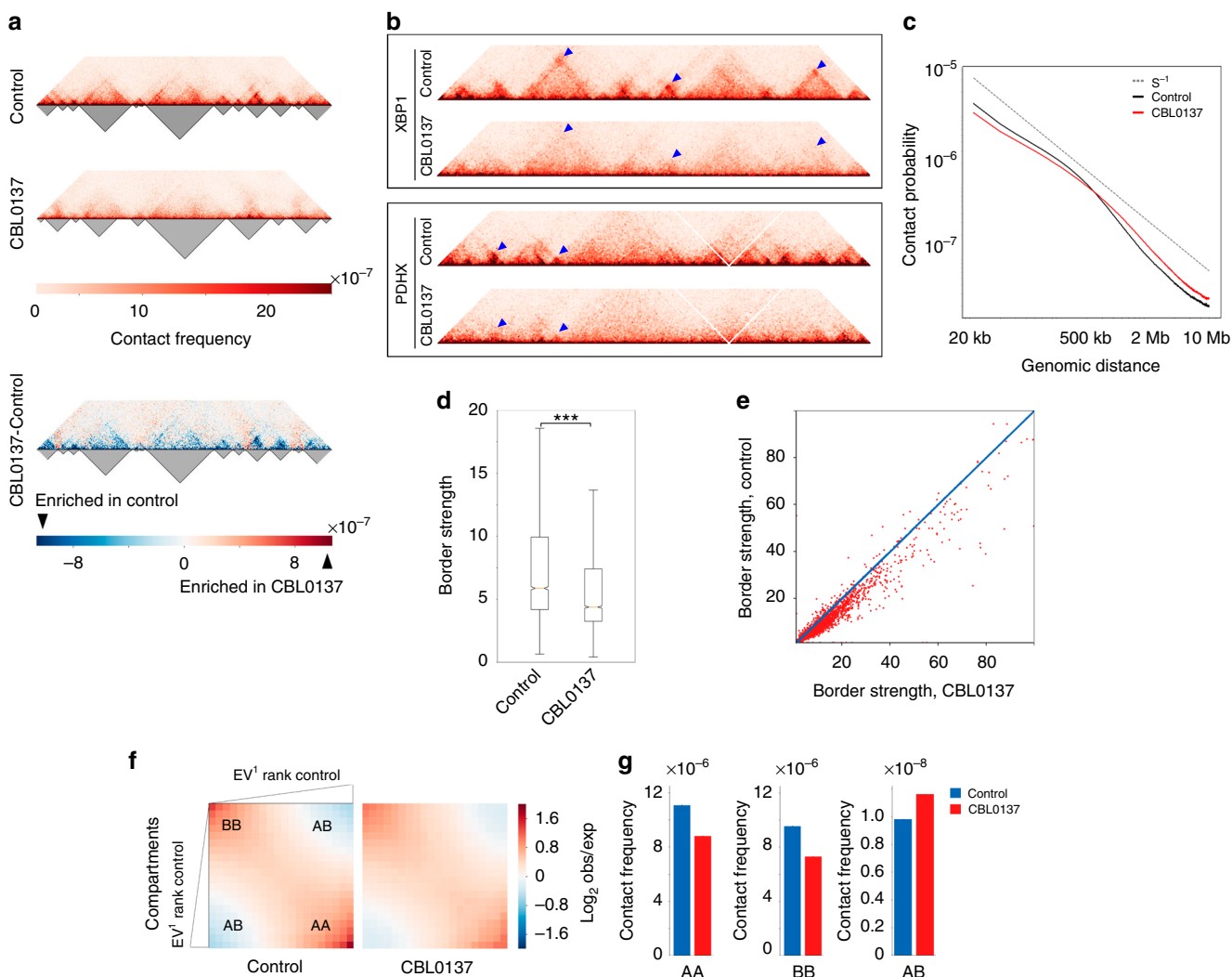

**Fig. 3** CBL0137 disturbs higher-order chromatin packaging. **a** HiC-map (heatmap) for a representative 10 Mb segment of chromosome 8 from control (upper panel) or CBL0137-treated HT1080 cells (middle panel). Heatmaps were normalized by the total number of sequencing reads, binned at a 20 kb resolution and iteratively corrected. TADs were annotated using the Lavaburst package (dark triangles below heatmaps). The heatmap in the lower panel shows the interactions enriched or depleted in CBL0137-treated cells compared to control cells. **b** Heatmaps showing the effect of CBL0137 on distant chromatin interaction patterns within the 4 Mb-long genomic segments centered at genes *XBP1* (upper panel) and *PDHX* (lower panel). Note the partial disruption of the TADs, and the decrease in the intensities of signals arising because of chromatin looping (blue triangles) in CBL0137-treated cells. **c** Scale plots showing the dependency of contact probability on genomic distance in control (black line) and CBL0137-treated (red line) cells. The dotted line indicates the contact probability $P(s) = s^{-1}$. **d** Box plots demonstrating the significant (P-value = $3.47 \times 10^{-5}$, N = 4862, t-test) decrease in the TAD border strength caused by CBL0137 treatment. Horizontal lines represent the median; upper and lower ends of boxplot show the upper and lower quartiles, the whiskers indicate the upper and lower fences. **e** Scatter plot demonstrating the value of TAD border strength in control and CBL0137-treated cells for each of the TAD borders annotated in the control cells. The location of most of the experimental points below the diagonal shows that the strength of the major part of the TAD borders decreased as a result of CBL0137 treatment. **f** Saddle plots displaying the extent of compartmentalization genome-wide in control and CBL0137-treated cells. **g** Frequencies of distant contacts within and between the A and B compartments in control and CBL0137-treated cells. Source data of Fig. 3c–e are provided in a Source Data file

approximately 24% of the downregulated genes belonged to the group of so-called essential genes necessary for cell survival[41] whereas only 3% of the upregulated genes belonged to this group (Supplementary Fig. 7).

There is no conclusive evidence that transcription strongly affects the 3D genome organization in vertebrates (reviewed in ref. [42]). Nevertheless, we investigated whether the effects of curaxins on the spatial organization of the genome are the consequences of massive transcriptional repression. We analyzed curaxin-induced changes in gene deserts that we defined as extended (> 500 kb) genomic regions that lack known genes and, consequently, are transcriptionally inactive (Fig. 5a). Visual

analysis of the Hi-C data for particular gene deserts clearly showed that CBL0137 compromised higher-order chromatin structure irrespective of the initial transcriptional activity of the region (Fig. 5b). This conclusion was supported by genome-wide analysis showing that the average border strength of the TADs located in the gene deserts was indistinguishable from that calculated for all TADs in the CBL0137-treated cells (Fig. 5c).

**CBL0137 partially depletes CTCF from its binding sites.** In vertebrates, CTCF, cohesin, and condensin almost exclusively maintain spatial genome organization[5]. In an attempt to uncover

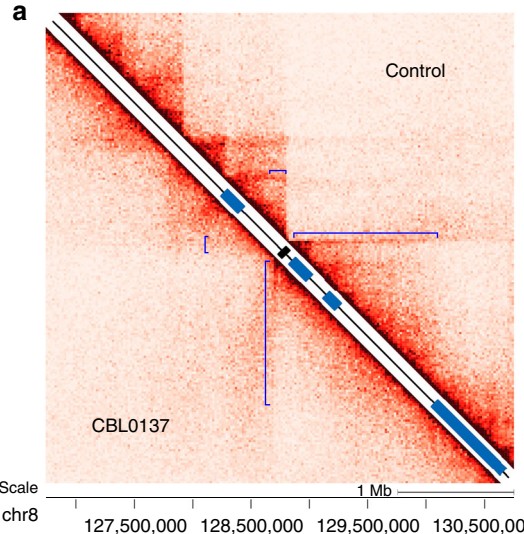

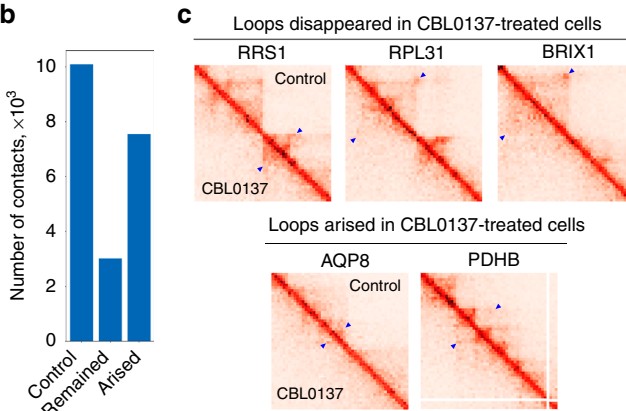

**Fig. 4** CBL0137 disturbs the chromatin loops that juxtapose distant genomic elements. **a** Hi-C maps showing that CBL0137 treatment suppressed the spatial interactions of the *MYC* gene (black rectangle) with distant enhancers (blue rectangles, presented according to ref. [12]). The regions of the Hi-C maps representing the spatial juxtaposition of *MYC* and distant enhancers are shown by blue square brackets. The size of the genomic region is 4 Mb. **b** Bar plots showing the total number of annotated distant (40 kb or longer) spatial contacts in control cells, the number of distant contacts preserved after CBL0137 treatment, and the number of spatial contacts originated de novo as the result of CBL0137 treatment. **c** Representative sections of HiC-maps showing chromatin loops demolished by CBL0137 treatment and chromatin loops originated de novo as the result of CBL0137 treatment. The signals from the loop bases are at the top of the blue triangles. The 1 Mb-long regions presented on the heatmaps are centered around genes indicated above the maps

the mechanisms underlying the effects of CBL0137 on chromatin structure, we examined whether CBL0137 affected these architectural factors. Treatment of HT1080 cells with CBL0137 for 6 h did not alter the protein levels of CTCF or the subunits of cohesin and condensin complexes (Rad21 and SMC2, respectively) (Fig. 6a). However, CBL0137 treatment of HT1080 cells led to a redistribution of CTCF, but not cohesin or condensin, from the fraction of proteins strongly associated with chromatin (Fig. 6b). This finding might reflect CTCF dissociation from its binding sites upon curaxin treatment. To test this assumption directly, we analyzed the genomic distribution of CTCF in control and CBL0137-treated cells using a chromatin immunoprecipitation-sequencing assay (ChIP-seq). In control HT1080 cells, about

45,000 CTCF-enriched peaks were mapped using the PePr computational approach[43], which is consistent with previously published data[44]. Moreover, the positions of the peaks were almost the same as those available from the ENCODE consortium (see the CTCF distribution in HT1080 (our data) versus HeLa S3 (ENCODE) in Supplementary Fig. 8). Being a crucial chromatin loop-organizing factor, CTCF was strongly enriched at the loop anchor regions in control HT1080 cells (Fig. 6c). Upon CBL0137 treatment, a portion of the CTCF peaks present in the control cells disappeared (Fig. 6c). Genome-wide, CTCF was depleted from approximately 40% of the initial peaks (Fig. 6d, e). The results suggest that curaxin-induced partial dissociation of CTCF from its binding sites might underlie the changes in 3D genome organization observed.

## Discussion

In recent years, efforts have been made to diversify cancer treatment strategies. Attention has been paid to novel therapies that exploit the synthetic lethality principle[45] or oncogene and general transcriptional addiction of cancer cells[17,18]. Here, we provided evidence that spatial genome organization may also be a target for anti-cancer chemotherapeutics. We demonstrated that the effective anti-cancer agents, curaxins, compromise EPC in vitro and in vivo, leading to preferential downregulation of enhancer- and SE-driven transcription. The curaxins exert selective cytotoxicity against transformed cells tested in a number of preclinical cancer models[22–26]. Despite extensive studies, their exact mode of action remains elusive, which is not surprising given that curaxins are small molecule compounds that intercalate into DNA[27] and, thus, are likely to exert different molecular effects. Compared to other known DNA intercalators, curaxins have the rare feature of not inducing DNA damage. In contrast to many known anti-cancer agents, curaxins inhibit the cleavage activity of topoisomerases and, thus, do not cause the formation of a cleavable complex or DNA breaks[27,28]. The most likely way for an intercalator to inhibit topoisomerases without forming DNA lesions is to block enzyme binding to DNA[46].

Indirect inhibition of the FACT histone chaperone is the most well-studied activity of the curaxins[27,28,47]. Curaxin binding to DNA induces nucleosome destabilization genome-wide[27,47], which, in turn, generates superhelical tension that cannot be relieved due to topoisomerase inhibition and induces B- to Z-DNA transitions[27]. During this process, multiple epitopes within the destabilized nucleosome become available for FACT binding. We have also shown that FACT can bind Z-DNA via the SSRP1 subunit[27]. As a result, all FACT complexes become tightly bound to chromatin in a process known as chromatin trapping of FACT or c-trapping[27]. Thus, growing evidence suggests that curaxins induce genome-wide changes in DNA and chromatin topology that result in both (i) the inability of some proteins to bind DNA efficiently and (ii) the trapping of some other proteins on chromatin.

Curaxins affect the physical properties of DNA, which results in increased rigidity of the chromatinized template and makes efficient EPC in vitro impossible. Modulation of chromatin fiber flexibility may be sufficient to modify the 3D organization of extended genomic segments and, thus, affect the EPC[48]. However, it is unlikely that the double helix alterations (increased rigidity, supercoiling, B- to Z-DNA transition) by themselves underlie the curaxin-induced changes in 3D genome observed here. Indeed, we showed that curaxins strongly compromise TAD structure and disrupt chromatin loops. The changes in chromatin structure observed were quite pronounced and, to a certain degree, mimicked the 3D genome alterations induced by CTCF depletion[49]. Thus, it was not entirely surprising to find that curaxins

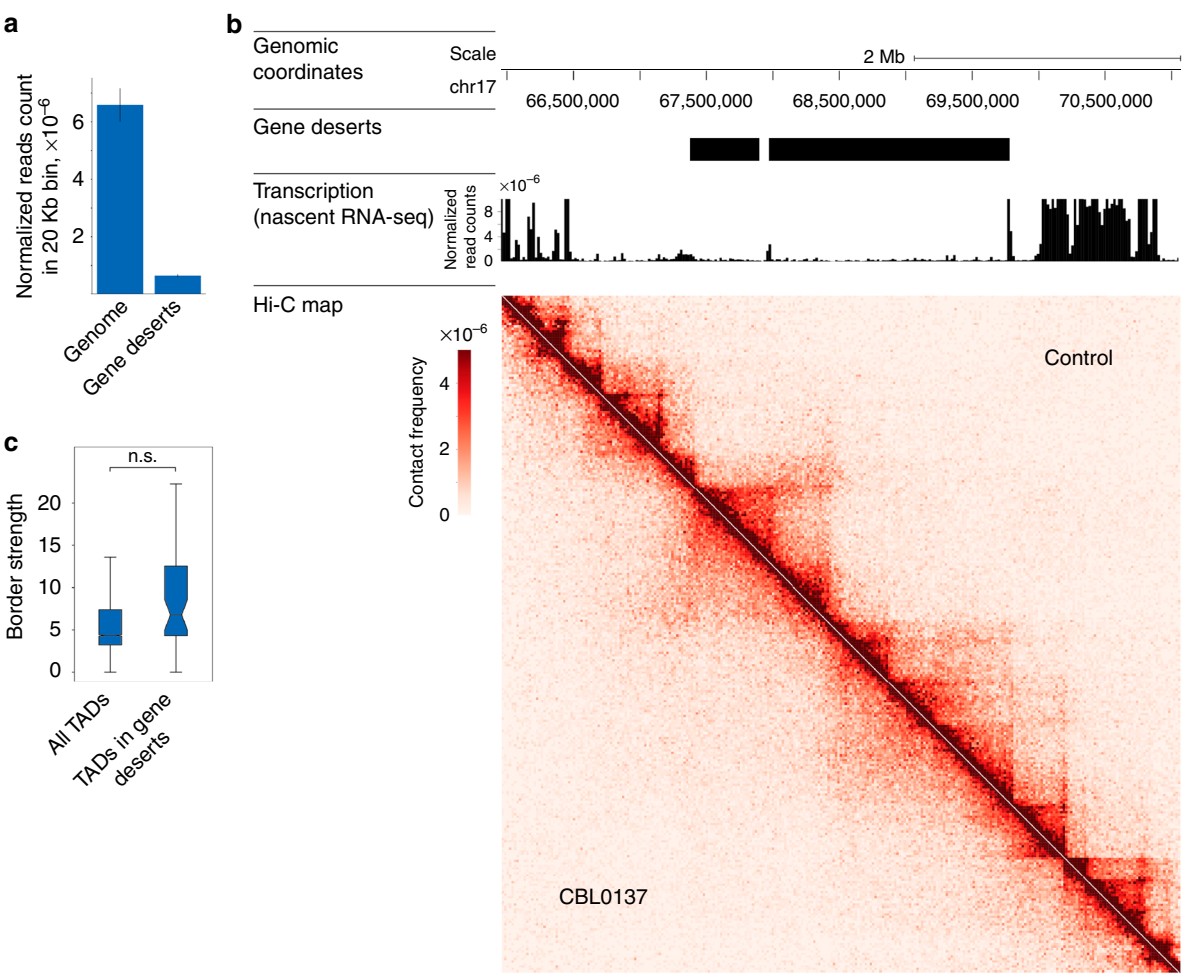

**Fig. 5** Effects of CBL0137 on spatial genome organization do not depend on transcription. **a** Bar plot shows the RNA normalized read count in a 20 kb-long bin calculated from nascent RNA-sequencing data from HT1080 cells. The average data are presented for all genomic bins ("genome") and bins located in gene deserts that we define as extended (> 500 kb) genomic regions lacking known genes. Error bars represent s.e.m. **b** Genomic segment from chromosome 17 containing a representative extended (~ 2 Mb) gene desert is shown. Corresponding genomic coordinates, mapped gene deserts, transcription levels (normalized reads count) across the region in intact HT1080 cells, and Hi-C maps for control and CBL0137-treated HT1080 cells are indicated. **c** Box plots showing no difference (*P*-value = 0.5137, *t*-test; n.s. – not significant) in the TAD border strength calculated for either all TADs mapped in CBL0137-treated HT1080 cells or TADs located in gene deserts. Horizontal lines represent the median; upper and lower ends of boxplot show the upper and lower quartiles, the whiskers indicate the upper and lower fences. Source data of Fig. 5a and c are provided in a Source Data file

could stimulate the partial dissociation of CTCF from its binding sites genome-wide. The inability of the curaxins to affect cohesin association with DNA likely reflects differences in the DNA-binding properties of CTCF and cohesin: while the first recognizes and physically binds specific DNA sequences, the second encircles DNA[42]. It is tempting to suggest that curaxin-induced DNA topology changes might modulate the affinity of CTCF for its binding sites. Of note, the affinity of CTCF for its binding sites strongly depends on the quality of the consensus sequence and its methylation status[50,51]. The inability of curaxins to remove CTCF from all of its genomic sites might be explained by their heterogeneity in terms of binding efficiency[51]. Non-promoter CTCF binding sites are often located in a linker region between precisely positioned nucleosomes[52] that makes them more vulnerable to the DNA topology changes induced by curaxins. It appears that CTCF largely mediates the effects of curaxins on the 3D genome; however, it is still questionable whether other transcription factors that are involved in spatial genome organization contribute to the effect.

In vertebrates, the contribution of active transcription to spatial genome organization has been debated for years (ref. [53] and references herein). However, very recently it was clearly shown that neither transcription nor replication is necessary for the re-establishment of the contact domains (chromatin loops) after they have been lost[54]. Our results corroborate this idea because the changes to the genome organization induced by curaxins were almost the same in transcriptionally active and silent (gene deserts) regions, which suggests that curaxins directly affect genome spatial organization thereby altering transcription.

In summary, we have shown that curaxins alter DNA topology leading to the inability of CTCF to bind efficiently to its cognate DNA sites. This effect on CTCF binding results in partial disruption of chromatin loops and in large-scale perturbations in the 3D genome organization, which alters EPC and leads to preferential downregulation of enhancer and SE-driven transcription. Thus, the curaxins are examples of drugs that target the spatial genome organization and have potential as an anti-cancer treatment.

## Methods

**Cells and CBL0137 treatment**. HeLa and MM1.S cells were obtained from ATCC. HT1080 cells were obtained from Dr. Andrei Gudkov (Roswell Park

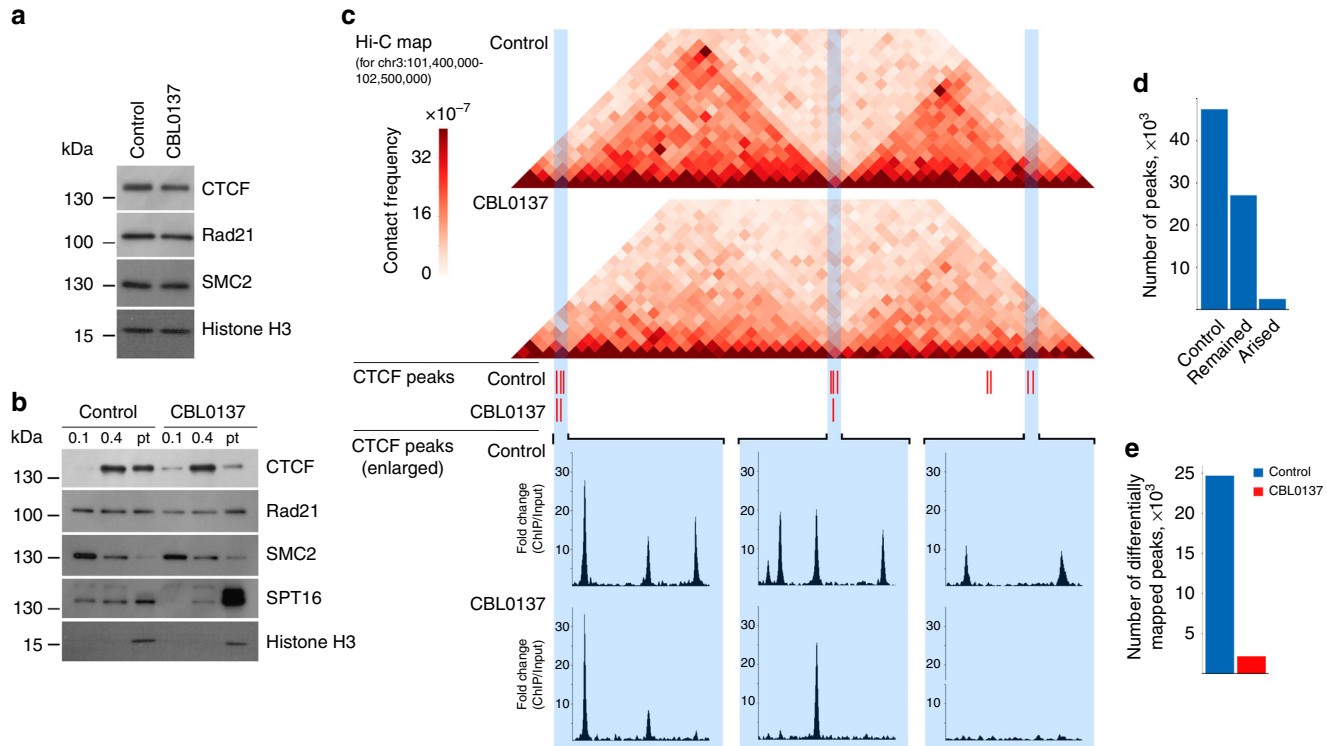

**Fig. 6** CBL0137 induces partial dissociation of CTCF from its binding sites. **a** Western blot analysis of CTCF, Rad21 (cohesin subunit), and SMC2 (condensin subunit) in nuclear extracts prepared from control and CBL0137-treated (3 μM, 6 h) HT1080 cells. Histone H3 was used as a loading control. **b** Western blot analysis of CTCF, Rad21, SMC2, SPT16 (FACT subunit), and histone H3 in different chromatin fractions from control and CBL0137-treated (3 μM, 6 h) HT1080 cells prepared by sequential extraction with 0.1 and 0.4 M NaCl. Pellet (Pt): insoluble chromatin fraction. **c** Representative 1.1 Mb-long genomic segment from chromosome 3 containing two distinct chromatin loop domains bordered by CTCF binding sites in control cells. Corresponding Hi-C maps, CTCF peaks determined using the PePr computational approach[43], and an enlarged view of the particular groups of CTCF peaks are shown for control and CBL0137-treated HT1080 cells. **d** Bar plots showing the total number of CTCF peaks mapped by PePr in intact HT1080 cells, preserved after CBL0137 treatment, and the newly formed CTCF peaks in CBL0137-treated cells. **e** Bar plots showing the number of CTCF differential binding sites identified in control and CBL0137-treated HT1080 cells. Analysis of differential binding sites was performed using PePr43, which showed the number of peaks mapped only in one of the two samples (either control or CBL0137-treated). Source data of Fig. 6a, b are provided in a Source Data file

Comprehensive Cancer Center) and were authenticated by short tandem repeat PCR to be 100% identical to HT1080 cells from ATCC (ATCC® CCL-121™). All cells were grown in Dulbecco's Modified Eagle's Medium supplemented with 10% fetal calf serum. Curaxin CBL0137 was provided by Incuron, LLC. Cells were exposed to CBL0137 (0.3–3 μM) for 6 h. The number of caspase-3/7-positive cells was measured using CellEvent Caspase-3/7 Green Detection Reagent (Invitrogen) according to the manufacturer's instructions.

**Reporter assays**. The NF-κB reporter construct was described previously[55]. All reporter constructs containing the minimal *MYC* promoter were cloned into the pGL3 plasmid as described[20]. Inserts for cloning were PCR amplified from MM1.S genomic DNA using the primer pairs provided in ref. [20]. All inserts were verified by sequencing. Plasmid DNA was transfected into cells using Lipofectamine 2000 (Invitrogen). Cells were split 24 h after transfection into replicate plates and treated with CBL0137 for 24–48 h. Reporter activity was measured using the BrighGlo kit (Promega).

**Expression analysis**. RNA extraction was performed using TRIzol (Invitrogen). cDNA was prepared using the iScript cDNA Synthesis kit (BioRad). Quantitative real-time PCR (RT-qPCR) was performed with mammalian Taqman probes (β-actin, ACTB [Hs01046520_m1]; MYC [Hs00905030_m1]) (Applied Biosystems) using the Applied Biosystems 7500 Real-Time PCR system following the manufacturer's protocol. The analysis was performed on triplicate PCR data for each biological duplicate normalized to β-actin.

Microarray hybridization was performed in the Roswell Park Genomics Shared Resource using the Illumina human Bead-ChIP array according to the manufacturer's instructions. Nascent RNA sequencing was done as described[28] using EU labeling of HT1080 cells for 15 min.

**Immunoblotting**. Cells were lysed in Promega Cell Culture Lysis Reagent. Immunoblotting was run as described[56]. The following antibodies were used:

c-MYC (9E10) (Santa Cruz Biotechnology, sc-40; 1:200), beta-actin (Sigma-Aldrich, A3854; 1:25,000), CTCF (Active Motif, 61311; 1:2000), Rad21 (Abcam, ab992; 1:2000), SMC2 (Cell Signalling, 5394; 1:2000), SPT16 (Abnova, MAB8018; 1:2000), and histone H3 (Abcam, ab1791; 1:2000). For chromatin fractionation experiments, cells were permeabilized in RSB buffer containing 10 mM Hepes-NaOH (pH 7.5), 1.5 mM MgCl$_2$, 0.5 mM EDTA, 10 mM KCl, 0.5% NP40, phosphatase, and protease inhibitors. After incubation at 4 °C for 10 min, cells were collected by centrifugation at 1000×g for 5 min. Cells were then incubated in RSB buffer containing 100 mM NaCl. After incubation at 4 °C for 10 min, the first soluble fraction (0.1 fraction) was separated by centrifugation at 10,000×g for 10 min. Cells were then incubated in RSB buffer containing 400 mM NaCl. After incubation at 4 °C for 1 h, the second soluble fraction (0.4 fraction) was separated by centrifugation at 8000×g for 10 min. The insoluble chromatin fraction (pellet) was then sonicated in RSB buffer at 50% amplitude for 30 s with a VirSonic 100 ultrasonic cell disrupter. Full size, uncropped scans or digital images of immunoblots are available in Supplementary Figs. 9 and 10.

**Identification of enhancers active in various cells**. A list of super-enhancers and typical enhancers for MM1.S cells was prepared according to Hnisz et al[12]. using ChIP-seq with anti-MED1 and anti-BRD4 antibodies. Enhancers were detected using ChIP-seq with H3K27Ac (H3 acetyl K27; Abcam, ab4729) in HT1080 cells incubated with or without 3 μM CBL0137 for 6 h. ChIP was performed with SimpleChIP Kit #9003 from Cell Signaling Technology according to the manufacturer's protocol. Library preparation and sequencing were done in the Roswell Park Genomics Shared Resource using Illumina NextSeq machine, which produced ~100 mln 75 bp PE reads per sample. Alignment was done using Bowtie 2[57]. Peak calling and annotation were performed using MACS[58].

**Preparation of template for in vitro analysis of EPC**. The construct used to assemble chromatin was described previously[30]. Proteins and protein complexes

were purified as described[30]. H1/H5-depleted chicken erythrocyte donor chromatin was prepared as described[59]. In vitro reconstitution of chromatin on linearized DNA templates was conducted using continuous dialysis from 1 M to 10 mM NaCl. Nucleosome positioning within the array was verified using the restriction enzyme sensitivity assay followed by primer extension[30]. Chromatin was assembled on EcoRI-linearized plasmids and digested with an excess of one of the following restriction enzymes: AluI, MspI or ScaI. Purified DNA was subjected to primer extension with Taq DNA polymerase using a radio-actively end-labeled primer, which anneals immediately upstream of the promoter[30].

**In vitro transcription-based analysis of the rate of EPC.** Templates preincubated with enhancer-binding protein NtrC and transcription machinery proteins[30] were incubated at the indicated concentrations of CBL0137 for 15 min at room temperature. The conditions for in vitro transcription were optimized for maximal utilization of the chromatin templates. Transcription was conducted as described[30]. Single-round transcription assays were conducted in transcription buffer (TB) containing 50 mM Tris-OAc (pH 8.0), 100 mM KOAc, 8 mM Mg(OAc)$_2$, 27 mM NH$_4$OAc, 0.7% PEG-8000, and 0.2 mM DTT with 1 nM linearized template, 10 nM core RNA polymerase, 300 nM σ[54], 120 nM NtrC, and 400 nM NtrB transcription factors. The reaction mixture was incubated for 15 min at 37 °C to form the closed initiation complex (RP$_C$). ATP was then added to the reaction mixture at a concentration of 4 mM, and the reaction was incubated at 37 °C for 2 min to form the open initiation complex (RP$_O$). Next, a 5-μl mixture of all four ribonucleotide-triphosphates (4 mM each) with 2.5 μCi of [α-$^{32}$P]-GTP (3000 Ci/mmol) and 2 mg/ml heparin was added to the reaction to start transcription and to limit it to a single round. The reaction was continued at 37 °C for 15 min and then terminated by adding phenol:chloroform (1:1). Labeled RNA was purified and analyzed by denaturing PAGE. The data were analyzed using a PhosphorImager (Bio-Rad). The rates of EPC were normalized to the values for histone-free DNA in the absence of CBL0137.

**Micrococcal nuclease digestion.** Chromatin templates (1 nM) preincubated with or without CBL0137 were digested by micrococcal nuclease (NEB; 800 gel units per reaction) in TB supplemented with 5 mM CaCl$_2$ for 2 min at room temperature. Reactions were stopped by adding EDTA to 17 mM. Digested samples were extracted with phenol/chloroform and labeled by PNK (NEB) in the presence of [γ-$^{32}$P]-ATP at 37 °C for 30 min. The digestion products were analyzed by PAGE.

**spFRET measurements.** The DNA and nucleosomes containing fluorescent labels (Cy3 and Cy5) for spFRET analysis were prepared as described[60,61]. The primer sequences used in the present work are presented below:

5'-ACACGGCGCACTGCCAACCCAAACG**T(Cy3)**CACCGGCACGAG-3'
5'-TAAGGCGAATTCACAACTTTTTGGC**T(Cy5)**AGAAAATGAGCT-3'

Labeled thymidines and fluorescent labels are shown in bold. spFRET measurements and analysis were performed as described[61].

The proximity ratio $E_{PR}$ was calculated as

$$E_{PR} = (I_a - 0.19 \times I_d)/(I_a + 0.81 \times I_d) \quad (1)$$

where $I_a$ and $I_d$ are Cy5 and Cy3 fluorescence intensities corrected for background. Factors 0.19 and 0.81 were introduced to correct for the contribution of Cy3 fluorescence in the Cy5 detection channel (spectral cross-talk).

The proximity ratios were calculated using 800–4800 signals from single nucleosomes for each measured sample and plotted as the relative frequency distribution. Each plot was fitted with a sum of two Gaussians to describe the two conformational states of nucleosomes (goodness of fit $R^2 = 0.91$–0.98). The mean maxima of peaks and standard errors were calculated from three independent experiments. The fractions of nucleosomes in the different states were estimated as the areas under the corresponding Gaussian peaks normalized to the total area of a plot.

Samples for spFRET measurements were prepared in a buffer (10 mM Tris-HCl (pH 8.0), 0.5 mM EDTA, 150 mM KCl) and contained either labeled DNA (0.5 nM) or labeled mononucleosomes (0.5 nM) in the presence of a 3-fold excess of long chromatin. Nucleosomes or DNA were preincubated with or without CBL0137 (0.5 μM) for 5 min at room temperature and measured under a microscope for 10 min. The reproducibility of the results was verified in three independent experiments.

**Generation and analysis of Hi-C libraries.** For Hi-C analysis, HT1080 cells were treated with 3 μM CBL0137 for 6 h. Hi-C was performed in two biological replicates using the DpnII restriction endonuclease as described[1,62]. Each Hi-C library was sequenced using paired-end sequencing on an Illumina HiSeq 3000 in two technical replicates; reads for each biological replicate were pooled and mapped to the human genome (version hg19) using hiclib[63] (https://bitbucket.org/mirnylab/hiclib/). Reads mapped in close proximity to the DpnII restriction sites (5 bp), reads mapped on the same fragment, and possible PCR duplicates were eliminated. Resulting pairs were binned into 20 kb genomic windows. As two biological replicates demonstrated a high correlation (Pearson's $r = 0.92$, for control

replicates, Pearson's $r = 0.90$ for CBL0137-treated replicates), we combined them and obtained ~150 million sequenced ligation junctions per control or CBL0137-treated cells after all filtration steps. Statistics of the Hi-C libraries sequencing and mapping, as well as the results of specialized Hi-C reproducibility tests, are presented in Supplementary Data 2. The combined contact maps were iteratively corrected[63] using cooler (https://github.com/mirnylab/cooler), and normalized by the total number of sequencing reads. TADs were annotated using the Lavaburst package (https://github.com/mirnylab/lavaburst), which provides a set of dynamic programming algorithms to assess an ensemble of TAD segmentations derived from a TAD scoring function. We used its optimal segmentation finder, which is based on the Armatus algorithm[64] using the TAD scoring function from that study (γ-parameter 0.3). The algorithm finds the global TAD segmentation of a contact map having the highest aggregate score.

Chromatin compartments were annotated using principal component analysis as described[1]. Briefly, on each map, we performed Principal Component Analysis, and the first component was taken. Per convention A/B-compartments were assigned by GC-content such that the A-compartment had a higher GC content than the B-compartment.

Saddle plots were generated as described[65]. We used the observed/expected Hi-C maps, which we calculated from the 20 kb iteratively corrected interaction maps of cis-interactions by dividing each diagonal of a matrix by its chromosome-wide average value. In each observed/expected map, we rearranged the rows and the columns in the order of increasing eigenvector value that was calculated for the control matrices). Finally, we aggregated the rows and the columns of the resulting matrix into 20 equally sized aggregated bins to obtain a compartmentalization plot (saddle plot).

Annotation of promoter-enhancer interactions was performed with PSYCHIC[39] (https://github.com/dhkron/PSYCHIC) using the parameter "shuffle" as the TAD initialization method. The input domains consisted of the TADs annotated using the Lavaburst package.

TAD border strength was calculated as the ratio between the sum of interactions inside the TADs to the sum of the interactions between a pair of TADs.

**CTCF ChIP-seq and data analysis.** ChIP-seq was performed with an anti-CTCF antibody (Active Motif, 61311) as described[66,67] for two biological replicates. ChIP samples were prepared for next-generation sequencing using a NEBNext Ultra II DNA library prep kit for Illumina (New England Biolabs). Libraries were sequenced on the Illumina NextSeq 550 and resulted in around 60 million 75-bp single-end reads per sample. Reads for each biological replicate were mapped to the human genome (version hg19) using Bowtie2[57] (version 2.2.3) with the '—very-sensitive' preset. Non-uniquely mapped reads were filtered using 'XS:i' flag. The resulting sam-files were sorted with possible PCR and optical duplicates filtered using Samtools[68] (version 1.5). Peaks were called using PePr[43] (https://github.com/shawnzhangyx/PePr) with a p-value cutoff of 0.05 and a sliding window size of 100 bp. Differential peak calling was performed with PePr[43] (https://github.com/shawnzhangyx/PePr) using the --diff option and intra-group normalization. The bigWig files were generated using deepTools[69] (version 2.0). For each biological replicate, the bigwig file was generated as the ratio of ChIP signal to input with RPKM normalization and a bin size of 50 bp. Smoothed bigwig files were generated as normal bigWig with smooth length parameter = 3 bins.

To identify the number of CTCF peaks remaining after CBL0137 treatment, the peaks for control and CBL0137-treated cells were called using PePr as described above. The peaks from control and CBL0137-treated cells were intersected using Bedtools[70] (version 2.27.1). For each pair of intersected peaks, the percent of intersection (ratio between the number of intersected DNA base pairs to the length of the larger peak) was calculated. The peak was defined as remaining if its percent of intersection parameter was more than 50%.

**Gene deserts analysis.** To annotate the gene deserts, RefSeq plus strand intergenic regions were identified and filtered for the presence of known genes on the minus strand. The obtained pool of intergenic regions was filtered by size, and only regions of ≥500 kb were considered gene deserts. To calculate the nascent RNA signal in the gene deserts, we split the genome into 20 kb bins. For each bin, we calculated the RNA signal as the ratio between the number of reads in the bin to the number of all mapped bins.

**Reporting summary.** Further information on experimental design is available in the Nature Research Reporting Summary linked to this article.

## Data availability

All datasets reported in this paper are available at the Gene Expression Omnibus with accession numbers GEO: GSE122463, GSE117611, GSE117409, and GSE107633. All other relevant data supporting the key findings of this study are available within the article and its Supplementary Information files or from the corresponding authors upon reasonable request. The source data underlying Figs. 1a–e, 2c–f, 3c–e, 5a, 5c, and 6a, b and Supplementary Figs. 3, 4, and 5 are provided as a Source Data file. A reporting summary for this article is available as a Supplementary Information file.

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

## Acknowledgements

We would like to thank Catherine Burkhart from Burkhart Document Solutions for critical review and editing of the article. This work was supported by Incuron, Inc (K.V. G.), the National Cancer Institute (R01CA197967 to K.V.G., R21CA220151 to V.M.S., and P30CA016056 to Roswell Park Comprehensive Cancer Center), the National Institute of General Medical Sciences (R01GM119398 to V.M.S.), the Russian Science Foundation (17-74-20030 to O.L.K.), and the Russian Foundation for Basic Research (18-29-07001). The development and application of the spFRET approach was supported by Russian Science Foundation (14-24-00031).

## Author contributions

K.V.G., O.L.K., S.V.R. and V.M.S. conceived the study. A.K.G., A.K.V., A.S., A.V.F., Ar.V.L., Al.V.L., E.V.N., K.V.G., M.E.V., O.L.K., S.V.R. and V.M.S. performed the experiments and/or analyzed the data. K.V.G., O.L.K., S.V.R. and V.M.S. wrote the manuscript. All authors edited the manuscript.

## Additional information

**Competing interests:** K.V.G. obtained a research grant and consulting payments from Incuron, Inc. The remaining authors declare no competing interests.

