## [Peer Review File · Nature Communications]

Reviewers' comments:

Reviewer #1 (Remarks to the Author):

The manuscript explores the effects of curaxin, a presumed anticancer drug, on gene expression. Using a series of reporter assays or in vitro experiments, authors conclude that curaxin inhibits the interaction of distal enhancers with promoters. Authors then examine this possibility genome-wide by performing Hi-C in control and curaxin-treated cells.

The manuscript is potentially interesting but very preliminary at this point. Authors should address the significant issues raised in the following comments:

1. Page 2. "We have recently discovered a class of anti-cancer agents, curaxins, which suppress transcription of oncogenes and affect the chromatin structure⁷". This sentence gives the impression that reference 7 reports the identification of a new class of drugs. However, reference 7 describes FACT as a sensor of torsional stress. This manuscript shows that curaxin uncoils nucleosomal DNA and causes the accumulation of negative supercoils. It is surprising that the authors ignore these previously published results in the interpretation of the data presented in the current manuscript. If curaxin does indeed act at a local level by affecting DNA supercoiling, then the title of the manuscript suggesting an effect on 3D genome organization may be misleading.

2. Page 2. "The effect cannot be reproduced in transient transfection experiments with a reporter gene expressed under control of MYC minimal promoter, alone or supplemented with an enhancer (Fig. 1c) and thus should be attributed to a genomic context". Authors should consider the likely possibility that plasmids in a transient transfection experiment are nicked or linearized. If curaxin exerts its effect by affecting supercoiling, this may not happen in linearized plasmid DNA.

3. Page 2. "Taking into account the inability of CBL0137 to suppress activity of enhancers in transient transfection experiments when enhancers are placed close to the promoter (Fig. 1c), the strong effect of CBL0137 on transcription of genes controlled by remote enhancers in living cells suggests that it is not the activity of enhancer per se but long-distance enhancer-promoter communication that is affected by CBL0137". It may not be appropriate to make this conclusion for the reasons described in #2 above, since authors are comparing what happens in the intact locus to what happens to reporter plasmids in transient transfection experiments.

4. Figure 2f and page 4. The FRET experiments are difficult to evaluate, since reference 17 is not published in a Journal available on PubMed. FRET intensity is inversely proportional to the power of 6 of the distance between the fluorophores. It is difficult to believe that the two fluorophores present in the linker DNAs flanking a nucleosome can be sufficiently close. Authors should comment on this and offer additional evidence.

5. Figure 3 and page 5. Differences in the Hi-C maps observed in various panels of Figure 3 could be due to differences in library quality. Authors should show information on quality control steps for the different biological replicates in a supplemental table. In particular, it would help to know if the number of intra- and inter-chromosomal interactions as well as short range (less than 20 kb) versus long range (more than 20 kb) is the same for each replicate of each sample.

6. Figures 3a and 3b. It appears from these figures that the strong Hi-C signal at the subunits of the TADs disappears in cells treated with curaxin. This signal is an indication of the formation of a CTCF loop. Authors should perform ChIP-seq of CTCF in the control and treated cell to determine whether the loss of the loop is due to loss of CTCF protein at the anchors.

7. Figures 5f and 5g, and page 5. Differences in compartments, which correlate with active and silenced chromatin, may be due to changes in transcription. Authors emphasize changes in transcription of Myc, but curaxin has a more generalized effect on transcription. This issue is

important because cells are incubated in curaxin for 6 hours, which could result in the depletion of many RNAs and proteins, leading to the observed changes in 3D organization. Authors should demonstrate that the observed effects are not an indirect consequence of transcription changes causing depletion of proteins involved in 3D organization, rather than a direct effect of curaxin on this process.

8. Page 6. To analyze the effect of curaxin on enhancer-promoter interactions, authors use PSYCHIC, which is a recently published program whose name does not inspire a lot of confidence when one is looking for statistical significance in the outcome. Does PSYCHIC determine significant interactions in the Hi-C data before finding interactions between enhancers and promoters? Since the authors only obtained approximately 150 million contacts after merging replicates, which is quite low for a mammalian genome, it is possible that the resolution in determining enhancer-promoter interactions is very low. This is not mentioned in the description. Are the authors able to assign each interaction to a single enhancer-promoter pair? Are interactions between enhancers or between promoters also affected, or were all these considered together? A more conventional way of doing this analysis would be to identify significant interactions using FitHi-C, for example, and then examine the effect of curaxin on these interactions after appropriate normalization.

Reviewer #2 (Remarks to the Author):

The authors studied the mechanism of a previously found drug (CBL0137) on suppressing cancer cells. Through in vitro and in vivo studies, they proposed a model where the drug disrupts the interaction between super-enhancers (SE) and promoters, and causes global 3D genome alterations at the TAD, loop and compartment levels. The specific effect of the drug on down-regulating MYC may contribute to its anti-cancer effect.

Specific comments:

1. The manuscript is short and formatted as a "Brief communications" type, rather than a typical NC style. The authors should consider to rewrite and add more details to comply to the NC format.
2. The drug causes a lot of changes in the 3D genome. How are the TAD and compartment changes correlate with gene expression changes?
3. The bromodomain inhibitors also disrupt SE functions and repress the expression of nearby genes. How does CBL0137 compare to these drugs in terms of drug efficacy, mechanism and the effect on the 3D genome?

Reviewer #3 (Remarks to the Author):

Kantidze et al. report that the drug curaxin CBL0137 affects genome organization by reducing intra-TAD interactions, which in turn promotes decreased expression of enhancer-regulated genes. CBL0137 is able to destabilize nucleosomes and stabilize Z-DNA formation, as shown by the same authors in a previous work, which might cause the reduced enhancer-promoter communications observed both in the chromatinized system and in cells by Hi-C. Based on these findings the authors present CBL0137 as a novel type of epigenetic drugs affecting genome architecture.

Overall, the manuscript provides interesting insights about the drug CBL0137 and its effect on chromatin. If the authors adequately address the comments outlined below, I recommend consideration of the manuscript for publication.

Major comments:

1) Kantidze et al. claim that CBL0137 suppresses enhancer-dependent transcription of many genes. By RNA-seq and WB the authors show that MYC expression is strongly decreased after drug treatment. In 2012, the groups of R. Young and D. Levens demonstrated that MYC is a “transcription amplifier”. It binds promoters and enhancers of active genes stimulating their expression exponentially. If MYC expression is strongly affected by CBL0137, all “MYC targets” (i.e. genes with open promoters bound by RNA polymerase) will be affected, which would explain the decreased expression of the high output promoters. Also, the group of R. Casellas has recently shown that reduction of MYC levels affects genome organization and promoter-enhancers interactions (Kieffer-Kwon KR, et al. *Molecular Cell* 2017). If the main point of the manuscript is to demonstrate the ability of CBL0137 to directly change the structure of the genome (as stated in the abstract) the authors need to rule out the possibility that this is a secondary effect of reduced MYC levels (for example, by testing the effect of CBL0137 on MYC independent genes).

2) Results in Figure 1C are a bit unclear. The expression of the reporter gene regulated by NF- κ B responsive element is affected by CBL0137 but not the one carrying the MYC promoter. The authors claim that this because of a “different genomic context”. I could not find information about the features of the MYC minimal promoter (in this paper or in the Young’s paper cited). Does the MYC minimal promoter contain the FUSE element? FUSE regulates the MYC promoter (Lui J, et al. *EMBO* 2006) and it is susceptible to negative supercoiling. Since CBL0137 preferentially binds underwound DNA, it can affect FUSE melting. Therefore the absence of the FUSE sequence might explain why the expression of the reporter gene is not affected. This possibility need to be investigated.

3) Kantidze et al. show that CBL0137 changes the structure and the properties of the linker DNA increasing the “fraction of nucleosomes with a larger distance between the linkers”. This finding supports a previous observation where the same authors demonstrated that CBL0137 destabilizes nucleosomes. Nucleosome destabilization should lead to a more open and flexible chromatin conformation and thus facilitate Enhancer-Promoter interactions. Follow this logic it is a bit unclear how the drug can promote decreased gene expression by favoring nucleosome destabilization. Is the chromatinized template characterized by Z-DNA forming sequences? Z-DNA could potentially form when transcription is turned ON and this might interfere with the EPC.

4) The presentation of the Hi-C data could be improved. For example, it would be nice to see a graph showing the effect of CBL0137 on segregation of A/B compartments as a function of gene density or gene expression (using the RNA-seq data already available). These analyses will support the overall message of the paper.

Minor points:

1) Introduction: Can the authors explain what are the “MYC family of genes”? Are these MYC-target genes? The authors might want to clarify the sentence.

2) Page 3 (2nd, 3rd line): According to Figure 1E or 1F (cited in the text), I think the comparison should be between super-enhancers and typical enhancers or presence vs. absence of enhancers. Please change the sentence.

3) Supplementary Figure 2B: Based on the micrococcal nuclease assay, the authors claim CBL0137 does not affect the nucleosome structure. I am not fully convinced by the figure. The overall quantity of DNA appears to be different in individual samples (see for instance the background of lanes 4 and 6). Thus, the question is: how do the authors check for equal loading among samples in the gel? This is particular important as differences are modest. Also, the authors might want to clarify which is the concentration that “strongly affects EPC”.

4) The authors use published RNA-seq data (Ref 27). Please add citation in the main text.

Point-to-point response to reviewers' comments:

Reviewer #1 (Remarks to the Author):

The manuscript explores the effects of curaxin, a presumed anticancer drug, on gene expression. Using a series of reporter assays or in vitro experiments, authors conclude that curaxin inhibits the interaction of distal enhancers with promoters. Authors then examine this possibility genome-wide by performing Hi-C in control and curaxin-treated cells. The manuscript is potentially interesting but very preliminary at this point. Authors should address the significant issues raised in the following comments:

1. Page 2. “We have recently discovered a class of anti-cancer agents, curaxins, which suppress transcription of oncogenes and affect the chromatin structure⁷”. This sentence gives the impression that reference 7 reports the identification of a new class of drugs. However, reference 7 describes FACT as a sensor of torsional stress. This manuscript shows that curaxin

uncoils nucleosomal DNA and causes the accumulation of negative supercoils. It is surprising that the authors ignore these previously published results in the interpretation of the data presented in the current manuscript. If curaxins does indeed act at a local level by affecting DNA supercoiling, then the title of the manuscript suggesting an effect on 3D genome organization may be misleading.

- We do agree that citation of manuscript by Safina et al (ref. 7 in the original version of the MS) in the sentence introducing curaxins was misleading. In the revised version of the MS we cite the appropriate article (ref. 21, Gasparian et al., 2011). As for the ability of curaxins to induce genome-wide changes in DNA and chromatin topology (including generation of local superhelical tension), we do not see why it contradicts our interpretation of the data. We think that these effects do contribute to the observed changes in 3D genome organization. The issue is now addressed in the Discussion section of the revised MS.

2. Page 2. “The effect cannot be reproduced in transient transfection experiments with a reporter gene expressed under control of MYC minimal promoter, alone or supplemented with an enhancer (Fig. 1c) and thus should be attributed to a genomic context”. Authors should consider the likely possibility that plasmids in a transient transfection experiment are nicked or linearized. If curaxin exerts its effect by affecting supercoiling, this may not happen in linearized plasmid DNA.

- We have isolated plasmid constructs from transfected cells using neutral Hirt extraction protocol (Arad, Biotechniques, 1998, PMID: 9591124). As shown by analysis of electrophoretic mobility most of the isolated plasmids were supercoiled and a significant portion remained supercoiled after exposure of cells to CBL0137. These observations are mentioned in the revised section of the MS: « To rule out a possibility that inability of CBL0137 to affect the activity of enhancer (and/or promoter) in these functional tests is due to damage and relaxation of reporter constructs, we extracted plasmids from transfected cells and analyzed them electrophoretically. The majority of plasmids preserved their supercoiled conformation in course of this transfection-extraction experiment». The photo of the gel is presented below for evaluation by the reviewer.

Neutral gel

Plasmids:

1. minMYC promoter
2. minMYC promoter + PDHX enhancer
3. minMYC promoter + IGLL enhancer

Figure legend:

Hela cells were transfected with the indicated plasmids and then treated with 3 uM of CBL0137 between 42 and 48 hrs after transfection. Plasmid DNA was isolated from these or mock transfected cells using Hirt extraction protocol (Arad, Biotechniques, 1998, PMID: 9591124). Separately plasmid DNA was linearized with single cutting restriction endonuclease to serve as a reference of plasmid size and state (supercoiled or relaxed or linearized). All samples were run in 1% agarose gel in neutral conditions. Based on the velocity of reference control plasmids most of DNA isolated from cells is in supercoiled state.

3. Page 2. *“Taking into account the inability of CBL0137 to suppress activity of enhancers in transient transfection experiments when enhancers are placed close to the promoter (Fig. 1c), the strong effect of CBL0137 on transcription of genes controlled by remote enhancers in living cells suggests that it is not the activity of enhancer per se but long-distance enhancer-promoter communication that is affected by CBL0137”.* It may not be appropriate to make this conclusion for the reasons described in #2 above, since authors are comparing what happens in the intact locus to what happens to reporter plasmids in transient transfection experiments.

- See the answer to the previous comment. We also would like to stress the attention that in the sentence cited by the reviewer we use the term “suggests” that acknowledges a possibility of other interpretations.

4. Figure 2f and page 4. *The FRET experiments are difficult to evaluate, since reference 17 is not published in a Journal available on PubMed. FRET intensity is inversely proportional to the power of 6 of the distance between the fluorophores. It is difficult to believe that the two fluorophores present in the linker DNAs flanking a nucleosome can be sufficiently close. Authors should comment on this and offer additional evidence.*

- For the fluorescent labels used in the experiment the Förster radius (the distance between the fluorophores providing 50% FRET efficiency, R_0) is 58 Å. Accordingly, changes in the inter-label distance can be measured with FRET in the ~ 30-90 Å range. The expected distance between the DNA linkers near the label positions is within this range. These positions of labels were selected among several different combinations tested by us. The paper (reference 17), although not in the PubMed, is available online:

<https://link.springer.com/article/10.3103/S0096392516020061>

The pdf file is attached to allow in-depth evaluation by the reviewers.

Moreover, the distances between the linkers have been studied by others using FRET previously (Toth et al., 2001; PMID:11389607). According to this work, inter-linker distance (end-to end distance) varies from 60Å for a 150-bp DNA template to 75Å for 170-bp DNA template organized into a nucleosome.

5. *Figure 3 and page 5. Differences in the Hi-C maps observed in various panels of Figure 3 could be due to differences in library quality. Authors should show information on quality control steps for the different biological replicates in a supplemental table. In particular, it would help to know if the number of intra- and inter-chromosomal interactions as wells as short range (less than 20 kb) versus long range (more than 20 kb) is the same for each replicate of each sample.*

- The requested information is presented in the Supplementary Table 2 of the revised version of the MS. Statistical characteristics measured show that Hi-C libraries prepared and analyzed in this study are of high quality according to (Belaghzal, Dekker, Gibcus, Methods, 2017, PMID: 28435001). Moreover, we have run two specialized tests to assess reproducibility of the generated Hi-C data, GenomeDISCO (Ursu et al., Bioinformatics, 2018, PMID: 29554289) and HiC-Spector (Yan et al., Bioinformatics, 2017, PMID: 28369339), which have displayed high reproducibility rate between replicates.

6. *Figures 3a and 3b. It appears from these figures that the strong Hi-C signal at the submits of the TADs disappears in cells treated with curaxin. This signal is an indication of the formation of a CTCF loop. Authors should perform ChIP-seq of CTCF in the control and treated cell to determine whether the loss of the loop is due to loss of CTCF protein at the anchors.*

- The experiment suggested by the reviewer has been performed. The results are presented in a new figure 6 of the revised MS and are explained in the corresponding paragraph (“CBL0137 induces partial depletion of CTCF from its binding sites”) of the Results section:

“CBL0137 induces partial depletion of CTCF from its binding sites. In vertebrates, CTCF, cohesin, and condensin almost exclusively maintain spatial genome organization⁵. In an attempt to uncover mechanisms underlying effects of CBL0137 on chromatin structure, we have analyzed whether it affects abovementioned architectural factors. We have shown that treatment of HT1080 cells with CBL0137 for 6 hours did not alter protein levels of CTCF, and subunits of cohesin and condensin complexes (Rad21 and SMC2, respectively) (Fig. 6a). Next, we analyzed the distribution of these proteins in different chromatin fractions obtained from control and CBL0137-treated HT1080 cells and found that CBL0137 treatment led to a redistribution of CTCF, but not cohesin and condensin, from the fraction of proteins strongly associated with chromatin (Fig. 6b). This might reflect the CTCF dissociation from its binding sites upon curaxin treatment. To test this assumption directly, we have analyzed the genomic distribution of CTCF in control and CBL0137-treated cells using chromatin immunoprecipitation-

sequencing assay (ChIP-seq). In control HT1080 cells, about 45 000 CTCF-enriched peaks were mapped using PePr computational approach 43 that is in a good agreement with the previously published data 44. Moreover, the positions of the peaks are almost the same to those available from ENCODE consortium (see an example of CTCF distribution in HT1080 (our data) versus HeLa S3 (ENCODE) cells on Supplementary Fig. 8). Being a chromatin loop-organizing factor of crucial importance, in control HT1080 cells CTCF is strongly enriched at loop anchor regions (Fig. 6c). Upon CBL0137 treatment, some portion of the CTCF peaks present in control cells disappears (Fig. 6c). Genome-wide, CTCF depletes from as many as ~40% of initially found peaks (Fig. 6d, e). The results suggest that curaxins-induced partial dissociation of CTCF from its binding sites may underlie the changes in 3D genome organization observed.”

7. Figures 5f and 5g, and page 5. Differences in compartments, which correlate with active and silenced chromatin, may be due to changes in transcription. Authors emphasize changes in transcription of Myc, but curaxin has a more generalized effect on transcription. This issue is important because cells are incubated in curaxin for 6 hours, which could result in the depletion of many RNAs and proteins, leading to the observed changes in 3D organization. Authors should demonstrate that the observed effects are not an indirect consequence of transcription changes causing depletion of proteins involved in 3D organization, rather than a direct effect of curaxin on this process.

- Using Western blot analysis we demonstrated that exposure of cells to CBL0137 (3 μ M, 6 h) did not cause a depletion of the major architectural proteins (CTCF, cohesin and condensin). These results are shown in Figure 6 of the revised MS. Following the reasoning of the reviewer we also considered a possibility that suppression of transcription *per se* can affect the 3D genome organization. This possibility was excluded by demonstration that the effect of CBL0137 on 3D genome organization was equally pronounced in transcribed areas and in gene deserts (Fig. 5 of the revised MS).

8. Page 6. To analyze the effect of curaxin on enhancer-promoter interactions, authors use PSYCHIC, which is a recently published program whose name does not inspire a lot of confidence when one is looking for statistical significance in the outcome. Does PSYCHIC determine significant interactions in the Hi-C data before finding interactions between enhancers and promoters? Since the authors only obtained approximately 150 million contacts after merging replicates, which is quite low for a mammalian genome, it is possible that the

resolution in determining enhancer-promoter interactions is very low. This is not mentioned in the description. Are the authors able to assign each interaction to a single enhancer-promoter pair? Are interactions between enhancers or between promoters also affected, or were all these considered together? A more conventional way of doing this analysis would be to identify significant interactions using FitHi-C, for example, and then examine the effect of curaxin on these interactions after appropriate normalization.

- There are indeed various algorithms for annotating spatial contacts between distant genomic elements. We preferred PSYCHIC because this algorithm is focusing on identifying intra-TAD contacts. We explained this in the revised MS: “In contrast to other related techniques like HiCCUPS⁴ and Fit-Hi-C⁴⁰, PSYCHIC-mediated annotation of promoter-enhancer interactions is TAD-specific³⁹. Thus, the data obtained by PSYCHIC are generally more accurate and is not skewed by TAD boundary elements³⁹”.

We were not able to assign each interaction to a single enhancer-promoter pair and did not aim to discriminate enhancer-promoter, enhancer-enhancer and promoter-promoter interactions. This may be a subject of another study. Here our aim was rather to show that CBL0137 exert a strong general effect on the 3D genome.

Reviewer #2 (Remarks to the Author):

The authors studied the mechanism of a previous found drug (CBL0137) on suppressing cancer cells. Through in vitro and in vivo studies, they proposed a model where the drug disrupts the interaction between super-enhancers (SE) and promoters, and causes global 3D genome alterations at the TAD, loop and compartment levels. The specific effect of the drug on down-regulating MYC may contribute to its anti-cancer effect.

Specific comments:

1. The manuscript is short and formatted as a "Brief communications" type, rather than a typical NC style. The authors should consider to rewrite and add more details to comply to the NC format.

-The revised MS is written according to Nature Communication format.

2. *The drug causes a lot of changes in the 3D genome. How are the TAD and compartment changes correlate with gene expression changes?*

- CBL0137 strongly suppress transcription, especially transcription of genes controlled by enhancers and super-enhancers (Fig. 1d-e of the revised MS). It is, however, difficult to discriminate the effect caused by changes in 3D genome from other effects related, for example, to trapping of chromatin remodeler FACT. On the other hand, our data strongly suggest that effect of CBL0137 on 3D genome is direct as (i) it can be observed in both transcribed and non-transcribed areas (Fig. 5 of the revised MS) and (ii) it is not due to the depletion of major architectural proteins that might be caused by transcription suppression (Fig. 6 of the revised MS)

3. *The bromodomain inhibitors also disrupt SE functions and repress the expression of nearby genes. How does CBL0137 compare to these drugs in terms of drug efficacy, mechanism and the effect on the 3D genome?*

- The most extensively characterized BETi is JQ1, an inhibitor of BRD4 (20871596) that inhibits BRD4 by preventing formation of a Mediator/BRD4-dependent phase-separated condensate necessary for activation of enhancer-driven transcription {Sabari, 2018 #8064;Cho, 2018 #8063;Boija, 2018 #8062}. Consequently it inhibits the action of enhancers *per se* rather than enhancer-promoter communication. In contrast, our results show that curaxins (CBL0137) do not significantly affect the action of enhancers *per se* but strongly compromise enhancer-promoter communication. Although this molecular mechanism differs substantially from that of BETi, the resultant cancer toxicity seems to depend (at least in part) on downregulation of *MYC* and its targets. This point of view is indirectly supported by several preclinical studies showing that, as well as BETi, curaxins display enhanced selective cytotoxicity especially in *MYC*-driven cancers (Carter et al., *Sci Transl Med*, 2015, PMID: 26537256; Zhang et al., *Cell Rep*, 2017, PMID: 28329685). However, it is difficult to directly compare BETi and curaxins because their molecular modes of action (initial molecular targets) are extremely different (proteins and DNA, consequently). That is why we have refrained from direct comparison of these small molecule compounds in the MS. Moreover, there is no any genome-wide data on BETi (or genetic downregulation of BRDs) influence on 3D genome; though this question deserves to be investigated in a separate study.

Reviewer #3 (Remarks to the Author):

Kantidze et al. report that the drug curaxin CBL0137 affects genome organization by reducing intra-TAD interactions, which in turn promotes decreased expression of enhancer-regulated genes. CBL0137 is able to destabilize nucleosomes and stabilize Z-DNA formation, as shown by the same authors in a previous work, which might cause the reduced enhancer-promoter communications observed both in the chromatinized system and in cells by Hi-C. Based on these findings the authors present CBL0137 as a novel type of epigenetic drugs affecting genome architecture. Overall, the manuscript provides interesting insights about the drug CBL0137 and its effect on chromatin. If the authors adequately address the comments outlined below, I recommend consideration of the manuscript for publication.

Major comments:

1) Kantidze et al. claim that CBL0137 suppresses enhancer-dependent transcription of many genes. By RNA-seq and WB the authors show that MYC expression is strongly decreased after drug treatment. In 2012, the groups of R. Young and D. Levens demonstrated that MYC is a “transcription amplifier”. It binds promoters and enhancers of active genes stimulating their expression exponentially. If MYC expression is strongly affected by CBL0137, all “MYC targets” (i.e. genes with open promoters bound by RNA polymerase) will be affected, which would explain the decreased expression of the high output promoters. Also, the group of R. Casellas has recently shown that reduction of MYC levels affects genome organization and promoter-enhancers interactions (Kieffer-Kwon KR, et al. Molecular Cell 2017). If the main point of the manuscript is to demonstrate the ability of CBL0137 to directly change the structure of the genome (as stated in the abstract) the authors need to rule out the possibility that this is a secondary effect of reduced MYC levels (for example, by testing the effect of CBL0137 on MYC independent genes).

- In Fig. 5 of the revised MS we present the evidence that CBL0137 strongly affects the 3D genome organization even in non-transcribed areas (gene deserts). Additionally, we demonstrate that exposure of cells to CBL0137 does not cause a depletion of major architectural proteins known to be essential for the establishing and maintaining of 3D genome (Fig. 6 a,b of the revised MS). These two lines of evidence disagree with a supposition that changes of 3D genome

organization in cells exposed to CBL0137 originate as a consequence of transcription suppression and strongly support a conclusion that effect of CBL0137 on 3D genome is direct.

2) Results in Figure 1C are a bit unclear. The expression of the reporter gene regulated by NF-KB responsive element is affected by CBL0137 but not the one carrying the MYC promoter. The authors claim that this because of a “different genomic context”. I could not find information about the features of the MYC minimal promoter (in this paper or in the Young’s paper cited). Does the MYC minimal promoter contain the FUSE element? FUSE regulates the MYC promoter (Lui J, et al. EMBO 2006) and it is susceptible to negative supercoiling. Since CBL0137 preferentially binds underwound DNA, it can affect FUSE melting. Therefore the absence of the FUSE sequence might explain why the expression of the reporter gene is not affected. This possibility need to be investigated.

- We do not claim that effects the CBL0137 exerts on transcription driven by NF-kB-regulated promotor and MYC minimal promotor are due to a different genomic context. We claim that effects CBL0137 exerts on transcription driven by minimal *MYC* promoter (supplemented or not supplemented with an enhancer) in transfected construct and *MYC* promoter in normal genomic position are due to a different genomic context. In the experiment presented in Fig 1 we used NF-kB construct as a positive control simply to verify that in this experiment we can reproduce previously observed suppression of NF-kB-dependent promoter by CBL0137.

There is no FUSE element in minimal *MYC* promoter. FUSE is 1.7kb upstream to TSS. To our knowledge, there is also no data that CBL0137 preferentially binds underwound DNA. The possibility that inhibition of genomic *c-MYC* gene expression in cells is due to the effect of CBL0137 on FUSE element solely is excluded by the fact that CBL0137 also inhibited expression of *MYCN* and translocated *MYC* (in MM1.S cells), both of which lack FUSE element. CBL0137 may have effect on FUSE, but this should be a subject of a separate study.

3) Kantidze et al. show that CBL0137 changes the structure and the properties of the linker DNA increasing the “fraction of nucleosomes with a larger distance between the linkers”. This finding supports a previous observation where the same authors demonstrated that CBL0137 destabilizes nucleosomes. Nucleosome destabilization should lead to a more open and flexible

chromatin conformation and thus facilitate Enhancer-Promoter interactions. Follow this logic it is a bit unclear how the drug can promote decreased gene expression by favoring nucleosome destabilization. Is the chromatinized template characterized by Z-DNA forming sequences? Z-DNA could potentially form when transcription is turned ON and this might interfere with the EPC.

- The chromatinized template used in this study was described previously (Polikanov&Studitsky, Methods Mol Biol, 2009, PMID: 19378187). There is no Z-DNA forming sequences in this template. Furthermore, the test system utilizing this chromatinized template does not use transcription through nucleosome (because after a single communication event and formation of the stable initiation complex nucleosomes are disrupted in the presence of heparin, see Fig. 2 legend). Therefore the effects of CBL0137 on transcription of the chromatinized templates observed *in vitro* are limited entirely to the enhancer-promoter communication step, as described in the original manuscript.

In living cells, CBL0137 exerts numerous effects on chromatin described in references 27-28. As mentioned in Discussion section all these effects can influence the long-range configuration of a chromatin fiber and thus compromise enhancer-promoter communication: “Modulation of chromatin fiber flexibility may be sufficient to modify the 3D organization of extended genomic segments and thus affect the EPC⁴⁸”. However, in the revised MS we also present the evidence that direct effect of CBL0137 on 3D genome organization may be due to displacement of CTCF from some of the binding sites (see Fig. 6 and related text in the Results section).

4) The presentation of the Hi-C data could be improved. For example, it would be nice to see a graph showing the effect of CBL0137 on segregation of A/B compartments as a function of gene density or gene expression (using the RNA-seq data already available). These analyses will support the overall message of the paper.

- Although it is clear that exposure of cells to CBL0137 decrease the segregation of A/B compartments it is not possible to present the graph proposed by the reviewer because there is no way to take into account the size of uninterrupted compartments. Of course, it is possible to present individual bins. However this will be counterintuitive because the neighboring bins should influence interactions at the compartment level.

Minor points:

1) Introduction: Can the authors explain what are the "MYC family of genes"? Are these MYC-target genes? The authors might want to clarify the sentence.

- In the first paragraph of the RESULTS section the phrase mentioning MYC gene family was modified as follows: "Expression of MYC family genes (c-MYC, NMYC, and LMYC²⁹) at both mRNA and protein levels is highly suppressed by CBL0137 in various cell lines, independently on whether they contained wild-type MYC gene locus or translocated MYC (Fig. 1a-b)"

2) Page 3 (2nd, 3rd line): According to Figure 1E or 1F (cited in the text), I think the comparison should be between super-enhancers and typical enhancers or presence vs. absence of enhancers. Please change the sentence.

- The Figure 1e was corrected (the comparison was made between genes controlled by enhancers (or super-enhancers in case of MM1.S) and genes lacking enhancers. The text of the first paragraph of results section was also corrected as follows: "Moreover, all genes regulated by enhancers (HT1080) or SEs (MM1.S) in these cells were inhibited by CBL0137 stronger and at lower concentrations than genes lacking remote enhancers (Fig. 1e-f)."

3) *Supplementary Figure 2B: Based on the micrococcal nuclease assay, the authors claim CBL0137 does not affect the nucleosome structure. I am not fully convinced by the figure. The overall quantity of DNA appears to be different in individual samples (see for instance the background of lanes 4 and 6). Thus, the question is: how do the authors check for equal loading among samples in the gel? This is particular important as differences are modest. Also, the authors might want to clarify which is the concentration that “strongly affects EPC”.*

- The apparent differences in DNA loading in Fig. S2B are primarily due to different labeling of MNase-digested DNA by the protein kinase. We carefully controlled the amount of material used in the experiment by measuring the absorbance at 260 nm (before digestion with MNase and controlled the extent of digestion by monitoring the ratios between the different bands in the gel. The ratios between the bands directly reflect the extent of the digestion. Quantitative analysis of the data in Fig. S2B is shown below and is included in the revised version of the figure:

The gel shown in A (corresponds to the original Fig. S2B) was quantified using a PhosphorImager. Amount of label present in each band was calculated as % of label present in all four bands (1-nucleosome etc.). The distribution of the label between the bands is very similar in all lanes, suggesting that the chromatin was digested by MNase to a similar extent.

Curaxins strongly affects EPC in vitro when present at concentrations of 1 or 2.5 uM (Fig. 2d). The concentrations are now described in the text. The corresponding phrase is modified as follows: “We found that addition of CBL0137 (1-2.5 μ M) causes a strong decrease in the yield of the transcript on the both chromatinized and free DNA model construct (Fig. 2c-d).”

4) The authors use published RNA-seq data (Ref 27). Please add citation in the main text.

- As requested, the citation (#28 in the revised version of the MS) has been added to the main text:

“To identify genes, which are inhibited by CBL0137 similarly to MYC, we analyzed the effect of CBL0137 on gene expression profiles in two human tumor cell lines, namely multiple myeloma MM1.S and fibrosarcoma cultured cells (line HT1080) using microarray hybridization and nascent RNA-sequencing²⁸.”

REVIEWERS' COMMENTS:

Reviewer #1 (Remarks to the Author):

The authors have addressed all my concerns. The new data, specially the effects on CTCF distribution, contribute to explain the observations and strengthen the conclusions of the manuscript.

Reviewer #2 (Remarks to the Author):

The authors have addressed my questions.

Reviewer #3 (Remarks to the Author):

Overall the revised manuscript has substantially improved and it is almost ready for publication in Nat. Communications. However,

- I would recommend to have the manuscript carefully checked by a native English speaker.
- There should be coherence in the wording. For example in page 4, what is a remote enhancer? To avoid confusion, please use always the same terms (enhancers and super enhancers if the authors decide to use those terms).
- Page 6. Meaningful apoptosis. It would be better to say "significant increase in apoptotic cells"
- Page 10. What do the authors mean with the term "non-promotor"?
- Figure 4a. What are the coordinate of the region shown in the figure?

Point-to-point response to reviewers' comments:

Reviewer #1 (Remarks to the Author):

The authors have addressed all my concerns. The new data, specially the effects on CTCF distribution, contribute to explain the observations and strengthen the conclusions of the manuscript.

Reviewer #2 (Remarks to the Author):

The authors have addressed my questions.

Reviewer #3 (Remarks to the Author):

Overall the revised manuscript has substantially improved and it is almost ready for publication in Nat. Communications. However,

- I would recommend to have the manuscript carefully checked by a native English speaker.

In order to improve manuscript's clarity and readability it was edited by a native English-speaking language editor - Catherine Burkhart from Burkhart Document Solutions.

- There should be coherence in the wording. For example in page 4, what is a remote enhancer? To avoid confusion, please use always the same terms (enhancers and super enhancers if the authors decide to use those terms).

We have made corrections to make manuscript more clear and accurate.

- Page 6. Meaningful apoptosis. It would be better to say "significant increase in apoptotic cells"

We have re-worded the phrase.

- Page 10. What do the authors mean with the term "non-promotor"?

This was a spelling error. We changed the word to the proper one – "non-promoter"

- Figure 4a. What are the coordinate of the region shown in the figure?

Scale bar representing coordinates of the genomic region shown in Figure 4a was inserted into the figure.